# Effect of *MSTN* Mutation on Growth and Carcass Performance in Duroc × Meishan Hybrid Population

**DOI:** 10.3390/ani10060932

**Published:** 2020-05-28

**Authors:** Weijian Li, Rongyang Li, Yinghui Wei, Xueqing Meng, Binbin Wang, Zengkai Zhang, Wangjun Wu, Honglin Liu

**Affiliations:** Department of Animal Genetics, Breeding and Reproduction, College of Animal Science and Technology, Nanjing Agricultural University, Nanjing 210095, China; 2017105018@njau.edu.cn (W.L.); 2018205007@njau.edu.cn (R.L.); wyh2017105016@njau.edu.cn (Y.W.); 2017105017@njau.edu.cn (X.M.); 2018205003@njau.edu.cn (B.W.); 2017105031@njau.edu.cn (Z.Z.); wangjunwu4062@163.com (W.W.)

**Keywords:** Meishan pig, *Myostatin*, gene knockdown, lean meat yield

## Abstract

**Simple Summary:**

*Myostatin* (*MSTN*) is a transcriptional growth factor that inhibits the development and growth of skeletal muscle. The *MSTN*-deficient animals display an increase in skeletal muscle mass known as double-muscling. Therefore, *MSTN* becomes an important target for improving lean meat production in livestock husbandry. There are many local pig breeds in China, but because of the slow growth, poor feed conversion, and low lean meat percentage and other unsatisfactory qualities, pure local breeds are rarely used on commercial farms. The objective of this study is to evaluate the effects of *MSTN* single allele mutation on carcass composition in Meishan crossbred pigs and demonstrate a way to increase lean meat yield while maintaining prolificacy and good meat quality of local pig crossbreeds. This has significant implications for the widespread use and conservation of local pig breeds in China.

**Abstract:**

The Meishan pig is a traditional Chinese native breed, known for its excellent reproduction performance that is widely used in commercial pig production through two-way or three-way crossbreeding systems. However, the lean meat yield of Meishan crossbred pigs is still very low and cannot meet the market demand. To evaluate the lean meat yield of Meishan crossbred pigs, six wild-type Meishan sows were artificially inseminated by using the *MSTN*^+/−^ Duroc boar semen in this experiment. Some reproductive performance-related traits of Meishan sows were recorded to ensure that semen from *MSTN* knockout Duroc boar did not affect offspring production, including total births, live births, sex, and litter weight. In total, 73 piglets were obtained and 63 were alive. Male to female ratio was close to 1: 1. because of factors such as disease, only 43 pigs were utilized, including 28 *MSTN* mutant pigs (*MSTN*^+/−^) and 15 *MSTN* homozygous pigs (*MSTN*^+/+^). We compared the growth performance and carcass performance of these full or half-sib populations and found that there were no differences between *MSTN*^+/−^ and *MSTN*^+/+^ genotypes for live animal measures including average daily gain (ADG), body dimensions, or ultrasonic measurement of fat thickness when pigs were harvested after 120 days of feeding. Conversely, the *MSTN*^+/−^ pigs had higher dressing percentage and lean meat percentage, lower level of carcass fat, larger longissimus muscle area, less percentage of skin and skeleton, thinner average backfat thickness, and lower intramuscular fat (IMF) content than *MSTN*^+/+^ pigs. In conclusion, the production of *MSTN*^+/−^ mutant progeny from Meishan females resulted in improved carcass composition, providing a feasible solution to improve the lean meat yield of Chinese local fat-type pig breeds.

## 1. Introduction

China is a large agricultural country with a long history. As an important part of agriculture, pig farming plays an important role in the development of agricultural animal husbandry. There are many varieties of indigenous swine breeds in China, which have the characteristics of good meat quality, excellent reproduction traits, good adaptability to extensive feeding and management, and the ability to utilize coarse grains. However, because of undesirable traits, such as slow growth, poor feed conversion, and low lean meat percentage, pure local breeds have been scarcely utilized on commercial farms in China. Crossbreeding is extensively used in pig production to increase the total efficiency of pig production. Given the outstanding characteristics of superior growth rate and lean percentage that boars of Duroc, Landrace, and Berkshire have [1], there is a rising trend to crossbreed Chinese native pigs with those from western countries to produce new breeds with the advantages of both. Meishan pigs are a famous local breed in China, known for its fecundity, greater fat deposition, and better meat quality [2]. Moreover, it is extensively utilized in commercial pig population by two-way or three-way crossbreeding systems. The most popular crossbreeding scheme utilizes Landrace × Meishan or Landrace × Meishan F1 sows bred to a terminal Duroc boar. Jiang et al. [3] compared Landrace × Meishan (LM), Duroc × (Landrace × Meishan) (DLM), two foreign crossbreeds Duroc × (Landrace × Yorkshire) (DLY) and PIC (an imported five-way crossbreed) and found LM had lowest percentage of carcass lean meat (46.82%), and the highest level of carcass fat and skin (40.40%) although Landrace crosses were generally superior. Because of the high carcass fat content of Meishan pigs, the lean meat yield of Meishan crossbred pigs is still low. In the past years, the main purpose of pig industry has been to increase the lean meat percentage of carcass [4]. At present, meat quality has become the main focus for pig production [5]. Therefore, it is necessary to improve the lean meat yield of Meishan crossbreeds based on the existing two-way or three-way crossbreeding systems.

The *MSTN* gene was first cloned in the mouse muscle cDNA library in 1997, also called growth differentiation factor-8 (GDF-8) [6]. The *MSTN* gene is homologous among different species and is mainly expressed in skeletal muscle of individual animals. Porcine *MSTN* gene is located on chromosome 15, including 3 exons and 2 introns. The *MSTN* gene is a cytokine that negatively regulates the growth of skeletal muscle. It can inhibit the formation and differentiation of muscle cells and impede the growth of skeletal muscles. Mutations or deletions can promote the proliferation and hypertrophy of muscle cells and muscle fibers [7,8]. In addition to affecting skeletal muscle development, the mutation of *MSTN* also inhibits fat deposition [9,10].

Within livestock industries, there has been great attention focused on *MSTN* inhibition for increasing lean tissue mass. According to the mouse *MSTN* gene sequence, the *MSTN* gene of cattle was cloned and found to have mutations in the exon of Belgian blue cattle and Piedmontese cattle. The muscle mass and lean meat yield increased greatly in these breeds because of the mutation, resulting in a double-muscled (DM) phenotype [11,12]. The appearance of the phenomenon, indicating that the *MSTN* gene is the main factor causing the phenotype of bovine DM. This has great potential for application in meat livestock (such as cattle, sheep and pig) with a large proportion of muscle. However, numerous production problems in homozygous *MSTN* mutant animals have been documented, including dystocia and reduced reproductive fitness [13]. According to some reports, heterozygous *MSTN* mutant animals were heavier at birth and at weaning, and their carcasses were leaner and more muscled than normal animals, demonstrating the benefits of incorporating single null *MSTN* alleles into breeding programs [14,15].

Recently, some studies have reported the production of *MSTN*-knockout pigs by using genome-editing technology combined with SCNT (somatic cell nuclear transfer). It is concluded that *MSTN*-knockout pigs have obvious double muscle phenomenon and the carcass lean meat yield is increased. The homozygous *MSTN* mutation was associated with early life mortality and calving problems [16,17]. These negative associations have limited the widespread exploitation of this phenotype for commercial gain, particularly in extensive farming systems. Besides, few information has been published on the growth and carcass performance of *MSTN* single allele mutant pigs. The purpose of this study, framed in a transgenic pig project focused on local pig breeds, is to more accurately evaluate the effect of *MSTN* single allele mutation on the growth and carcass composition and quality in Duroc × Meishan hybrid population.

## 2. Materials and Methods

### 2.1. Animal Ethics Statement

All the *MSTN*^+/−^ and *MSTN*^+/+^ pigs were fed with the same standard diet and raised under the same conditions. All experimental protocols related to animal work described in this study were reviewed and approved by the Institutional Animal Care and Use Committee of Nanjing Agricultural University (SYXK2011-0036, 6 December 2011).

### 2.2. Animals and Management

We crossed a Duroc *MSTN*^+/−^ boar with six wild-type Meishan pigs by artificial insemination to produce Duroc × Meishan pigs. Six adult Meishan sows were provided by a local livestock farm in Changzhou city (Jiangsu Province, China), the semen of *MSTN*^+/−^ Duroc pig was supplied by China Agricultural University. All operations of artificial insemination were performed by professionals. Because of factors such as disease, 28 *MSTN*^+/−^ (13 males, 15 females) pigs and 15 *MSTN*^+/+^ (8 males, 7 females) pigs were obtained after genotyping. Males were castrated at 4 ± 1 day of age. Barrows and females were mixed feeding and were kept in intensive commercial pig breeding conditions from birth to slaughter during the experiment. All pigs were fed twice a day with the same food, and had *ad libitum* access to water via nipple drinkers. The experimental diets were based on corn and soybean meal and were formulated with crude protein concentrations, trace minerals, vitamins to meet or exceed National Research Council [18] recommendations for the different growth phases (Table 1). Chemical analyses of the basal diet were carried out according to the methods of AOAC [19].

### 2.3. Traits Measurements

The basic reproductive performance of Meishan sows was counted, including total litter size at birth, alive litter size, sex, and litter weight. Total litter size at birth were measured as the total number of piglets born, including stillborn. Pigs were weighed within 12 h of birth for litter weight.

The feeding experiment of pigs was started at the age of 90 days and lasted for 120 days, the data of initial and final live weight, backfat depth on living pig and body size were measured in accordance with the rules for performance testing of breeding pig. Body size parameters measurements included body length, body height, chest girth, abdominal girth, hip girth, and cannon girth, were taken by using a measuring tape and modified caliper. Backfat depth was measured using ultrasound (Mylab Touch Vet, Esaote Ltd. Genoa, Italia) on the left side of the live pig at the 3rd to 4th last rib location. Then, all pigs were slaughtered to determine carcass composition according to the methods described by Xiao et al. [20]. Briefly, pigs were removed from feed the night before slaughter, transported in the morning 1 h to the abattoir, and allowed to rest for 2 h with access to water prior to slaughter. Pigs were electrically stunned (90 V, 10 s, and 50 Hz), exsanguinated, dehaired, and eviscerated. The head was removed and the carcass was longitudinally split. Hot carcass weight was recorded and used to calculate carcass yield. The left carcass was physically dissected into bone, muscle, and subcutaneous fat and skin, and each of the dissected tissue was weighed to the nearest gram. The average backfat thickness was measured in the midline on the thickest part of scapula, last rib, and last lumbar vertebrae, with a sliding caliper, and the skin thickness was measured at the 6–7 rib of the centerline of the carcass. The longissimus muscle area (LMA) was determined by tracing its surface area at the 10th rib and by using a planimeter (Planix 5.6, Tamaya Digital Planimeter, Tamaya Tecnics Inc., Tokyo, Japan).

The LMA between the 10th-rib and the first lumbar vertebra were evaluated for meat quality traits. The pH was measured using a pH star (Osaka, Japan) at 45 min and 24 h postmortem on the LMA according to the procedure described by Alonso et al. [21]. Color parameters were determined at 45 min after slaughter, by using a Minolta CR-300 colorimeter (Minolta Camera, Osaka, Japan) with an illuminant D65, a 0° standard observer, and a 2.5 cm diameter port/viewing area according to the procedure described by Miao et al. [22]. Drip loss was defined as the weight loss of a meat sample (50 g), placed on a flat plastic grid and wrapped in foil, after a storage time of 24 h in a refrigerator [22]. The analysis of intramuscular fat (IMF) content was measured according to the AOAC [19] procedures.

### 2.4. Genotype Assessments

The genotyping of the piglets was based on the protocol provided by China Agricultural University [23]. The *MSTN* mutant Duroc pig was produced by ZFN combined with SCNT technology. ZFN plasmid pairs PZFN1/PZFN2 were specifically designed to target exon 2 of porcine *MSTN* gene. Modification of the specific ZFN target sites is expected to result in the loss of *MSTN* function. In order to distinguish between wild-type and heterozygous type, genomic DNA was extracted from piglets and then was subjected to PCR to detect ZFN-targeted region of *MSTN* gene.

### 2.5. RNA Extraction and Real-Time Quantitative Polymerase Chain Reaction

Total RNA was extracted from LMA of *MSTN*^+/−^ and *MSTN*^+/+^ pigs using TRIzol (Invitrogen, Carlsbad, CA, USA) according to the manufacturer’s protocol. The transcription level of *MSTN* was measured via real-time PCR (RT-PCR). The purity and quantity of total RNA was measured by a NanoDrop 2000 spectrophotometer (Thermo Scientific, Wilmington, DE, USA) at 260 and 280 nm. The RNA was treated with DNase I (Takara Biotechnology Co. Ltd., Nanjing, China) to remove DNA and reverse transcribed to cDNA (10 μL reaction system for maximum use of 500 ng of total RNA) using a PrimeScript RT Master Mix kit (Takara Biotechnology Co. Ltd., Nanjing, China) following the instructions.

Primers of DEGs were designed by Primer 3 (http://bioinfo.ut.ee/primer3/) and are listed in Table 2 (*MSTN* NM_010834.3, *GAPDH* NM_001289726.1). The primer pairs used for detecting *MSTN*-total are located in Exon1 region (1546–1565) and the beginning region of Exon2 (3534–3552) respectively, and the primer pairs used for detecting *MSTN*-intact are located in the end region of Exon2 (3744–3766) and Exon3 (5903–5922). Since the deleted segment in mRNA is located 3605–3798, the primer set of *MSTN*-intact would amplify less fragment with mRNA samples from *MSTN*^+/−^ pigs.

Real-time quantitative reverse transcription polymerase chain reaction (qRT-PCR) was performed with AceQ qPCR SYBR Green Master Mix (Vazyme, Nanjing, China) in a reaction volume of 20 μL, containing 10 µL of SYBR Green Master Mix, 0.4 µL of each primer (10 µM), 0.4 µL of ROX Reference Dye II, 2 µL of cDNA, and 6.8 µL of sterilized doubled-distilled water. The cycling parameters are as follows: 95 °C for 5 min, followed by 40 amplification cycles, each at 95 °C for 10 s, then 60 °C for 30 s. All reactions were performed in triplicate for each sample. The glyceraldehyde-3-phosphate dehydrogenase (*GAPDH*) gene was used as the internal control to normalize the relative expression of genes. The gene expression levels were calculated using the 2^-∆∆CT^ value method.

### 2.6. Statistical Analyses

The effects of sex (barrow and female) and genotype (*MSTN*^+/+^ and *MSTN*^+/−^) on growth traits, body size traits, carcass composition traits, and meat quality traits were analyzed with GLM (general linear models) of SPSS ver.25.0 (SPSS Inc., Chicago, IL, USA). An analysis of covariance with the final body weight as covariant was implemented. Statistical significance of the expression level of *MSTN* between *MSTN*^+/+^ and *MSTN*^+/−^ pigs was determined by the Student t-test. The significance level was *p* ≤ 0.05 for all the measurements. Values in the text are expressed as means ± pooled SEM.

## 3. Results

### 3.1. Reproductive Performance

To investigate whether *MSTN* mutation in boar semen could affect offspring production, the reproductive performance of Meishan sows was recorded and is shown in Table 3, including litter size at birth (LS), alive litter size, sex, litter weight at pigs born alive (LW), and birth weight of pig born alive (BW). A total of 73 piglets were born and total alive litter size was 63. The litter size at birth (9–15) was normal for industry and the number of males and females were almost the same. The birth weight of the pigs born alive (0.96–1.18kg) was at a low level, and normal range in Meishan pigs.

### 3.2. Genotyping and MSTN Expression Detection

Because of the issues with disease in some of the available pigs, 43 pigs were utilized in the growth and carcass composition and quality assessment components of the study, including 21 barrows and 22 females. The results of progeny genotyping were shown in Figure 1. The specific ZFN target sites could be detected in hybrid *MSTN*-mutant progeny, while in wild-type offspring it could not be detected. After identification, there were 28 *MSTN*^+/^^−^ pigs (13 barrows and 15 females) and 15 *MSTN*^+/+^ pigs (8 barrows and 7 females). The detection of *MSTN* gene expression in muscle also confirmed the production of *MSTN*^+/^^−^ pigs. The mRNA sample from *MSTN*^+/+^ pigs could be successfully amplified by the primer sets of the *MSTN*-total and *MSTN*-intact (Figure 2). Compared with the *MSTN*^+/+^ pigs, the *MSTN*-total mRNA of *MSTN*^+/^^−^ pigs did not change significantly, but its *MSTN*-intact mRNA decreased significantly (Figure 2).

### 3.3. Body Size and Growth Performance

The growth performance of two groups was measured under the same feeding conditions to investigate the effects of *MSTN* mutation (Table 4). There was no significant difference in body weight and average daily gain between the two genotypes. The living body backfat thickness of *MSTN*^+/−^ pigs was thinner than *MSTN*^+/+^ pigs, but the difference was not significant. Direct measurements of body size traits indicated that both genotypes of pigs have no significant differences in body length, body height, chest girth, abdominal girth, cannon girth, and hip girth (Table 5).

### 3.4. Carcass Performance and Meat Quality

To investigate whether *MSTN* mutations affects carcass performance, we conducted slaughter experiments on two genotypes of pigs. As the results show, the *MSTN*^+/−^ pigs had better dressing percentage, greater percentage of lean in the carcass, lower level of carcass fat, larger longissimus muscle area, less percentage of skeleton and skin, and thinner average backfat thickness compared to *MSTN*^+/+^ pigs. No significant differences in skin thickness at 6/7 ribs (Table 6) were found. In meat quality analysis, only the IMF appeared to be significantly different between the two groups (Table 7). No significant differences in pH, color, and drip loss were found.

## 4. Discussion

In recent years, some studies had reported the production of *MSTN*-knockout pigs by using genome-editing technology combined with SCNT [23,24,25], and all of these *MSTN*-KO pigs had visible double muscling (DM) phenotype. Although DM animal such as Belgian Blue beef cattle had 20% more muscle mass on average, less bone, lower fat, the breed indeed had several disadvantages, in particular, the reproduction issues due to unusually heavy and bulky offspring and reduced reproduction tract [16]. Also Arthur et al. reported that the DM phenotype could reduce fertility and it had been suggested that DM embryos had a higher mortality rate [17]. Recently, Han et al. showed that *MSTN*-KO boars had no abnormalities in the reproductive organs. The semen color, odor, and pH also had no abnormalities [26]. Although the reproductive performance was not the focus of our research, we want to know whether *MSTN* mutations in boar semen could affect offspring production, which is related to the feasibility of the hybridization scheme. Therefore, we simply counted the reproductive performance of Meishan sows, which were consistent with other reports [27,28]. All these results showed that *MSTN* knockdown in boar did not significantly affect the birth of progeny. Thus, these *MSTN* heterozygous mutation pigs will have an obvious advantage for the livestock meat industry.

Sex has a greater impact on pig growth and carcass performance. Intact male pigs have improved G:F and carcass leanness compared to physically castrated barrows [29,30] but can produce boar taint, the unpleasant odor released when pork from some intact males is cooked. Consequently, male pigs destined for meat production have normally been surgically castrated shortly after birth to eliminate the risk of boar taint. Some research showed barrows had higher average daily feed intakes (ADFI) and average daily gains (ADG) than gilts from 75 to 116 kg BW, and higher ADFI than gilts from 116 to 124 kg [31]. However, inconsistent conclusions had also been reported that sex did not affect growth performance [32]. Carcasses produced by the barrows were heavier than those of the gilts, even though other studies reported that carcass weight and yield were not affected by sex [31,33]. Backfat thickness was affected, with intact females having a lower backfat thickness than males [34].

In brief, gender as a potential influencing factor, we cannot ignore its influence in this experiment. In the current results, there was no significant difference in growth performance and little influence in body dimension, carcass composition, and meat quality between sexes.

Earlier research showed that *MSTN*-KO Erhualian pigs and Meishan pigs had obvious muscular protrusion, wide back, and fuller rump compared with the wild-type control [23,25]. However, there was no detailed description of the body size of *MSTN*^+/^^−^ pigs compared with wild-type pigs. In this study, we did not observe the body shape differences in *MSTN*^+/^^−^ pigs and there was no significant difference between *MSTN*^+/^^−^ and *MSTN*^+/+^ pigs in body dimensions measurements. Both types of pigs showed similar growth status with similar ADG and body weight, which was similar to previous research in cattle [14]. Most importantly, there were large differences between the two groups in carcass performance, the *MSTN*^+/^^−^ pigs had better dressing percentage, higher percentage of lean carcass weight, lower level of carcass fat, larger longissimus muscle area, less percentage of skeleton and skin, and thinner average backfat thickness compared with *MSTN*^+/+^ pigs. The percentage of lean meat yield of *MSTN*^+/^^−^ barrows was 55.41%, approximately 9% greater than the corresponding *MSTN*^+/+^ barrows, and the fat percentage was reduced by 8.48%. These results indicated single allele *MSTN* mutation could increase the skeletal muscle mass and decrease fat content in Meishan crossbred pigs. Similar results had been previously reported. In homozygous and heterozygous *MSTN*-mutant Meishan pigs, the lean percentage nearly improved 12% and 4%, respectively, compared to wild-type pigs [23], and other parameters were also consistent with those found in the present study. Besides, *MSTN* not only negatively regulates the growth of skeletal muscle, but also affects the process of fat metabolism. The molecular mechanism by which fat in *MSTN*-mutant mice and *MSTN*-mutant Meishan pigs were reduced was mainly due to the browning of white fat, which led to an increase in fat consumption [35,36]. Meishan pigs are considered a famous Chinese pig breed for super-prolificacy because of its large litter size [37]. Like most of local Chinese breeds, they grow slowly and deposit huge amounts of body fat mass with low feed efficiency [38], and are extensively utilized in commercial pig population by two-way or three-way crossbreeding systems. The most popular crossbreeding scheme utilizes Landrace × Meishan or Landrace × Meishan F1 sows bred to a terminal Duroc boar. However, the lean meat yield of Meishan crossbred pigs is still low. According to numerous studies, the lean meat yield of 1/2 Meishan hybrid pigs is about 45% [3,39,40]. So there is still a lot of room for improvement in lean meat yield, and our research has great significance for this.

In meat quality analysis, only the IMF appeared to be significantly different between the two groups. Pig selection over the past few decades had mainly focused on increased lean meat production, however, a consequence had been a decrease in quality. One of the most important traits influencing the sensory characteristics of fresh pork was IMF content [41]. This had a positive effect on flavor, juiciness, tenderness, and overall acceptability of pork [42,43]. An IMF content of 2–3% was suggested to be optimal for eating quality [44]. However, IMF levels in the majority of modern commercial breeds had decreased below 1.5%, because lean meat content and IMF content were negatively correlated, the selection for increased lean efficiency had led to a decrease in IMF to levels below those recommended [45]. In our research, the *MSTN*^+/^^−^ pigs showed high lean meat yield and low IMF content compared with wild-type pigs, which was consistent with previous studies. However, the IMF content (2.66%) was within the ideal range of 2–3%, which could meet the demand for high taste quality products.

There were no differences in other traits such as meat pH and drip loss. The pH of pork is a major parameter that defines meat quality which affected both the technological and eating qualities of pork [46]. Some studies suggested that the local breeds have higher muscle pH (45 min). In this study, the two groups had pH (45 min) values, within normal scopes, which suggested that there were no characteristics of PSE (pale, soft, exudative) meats usually associated with pH (45 min) values lower than 5.9 [47]. None of the meat color values was significantly influenced by *MSTN* single allele mutation. The present results corroborated those of previous reports, which found that *MSTN* single allele mutation had no effect on meat color [48,49]. Water is the major meat constituent representing approximately 75% of the meat weight. It is an essential quality parameter, both for the industry and the final consumer. High water holding capacity (WHC) values might cause advantages both in processed meats for the industry and in the fresh meat appearance for the consumer [50].

## 5. Conclusions

Above all, we use the semen of a *MSTN*^+/^^−^ Duroc boar (supplied by China Agricultural University) to produce a certain number of wild-type and *MSTN*^+/^^−^ pigs by artificial insemination technology; knocking down *MSTN* in Duroc boar did not affect the generation of progeny. By evaluating the growth and carcass performance of these two groups of full or half sib populations under the same feed condition, we found that the *MSTN*-mutant pigs could grow normally and had the same body shape as the wild-type, the *MSTN*-mutant pigs had higher dressing percentage, higher percentage of lean carcass weight, lower level of carcass fat, less percentage of skeleton and skin, thinner average backfat thickness, and more ideal IMF content than *MSTN*^+/+^ pigs, which demonstrate a way of fast genetic improvement for local fat-type crossbreed.

## Figures and Tables

**Figure 1 animals-10-00932-f001:**
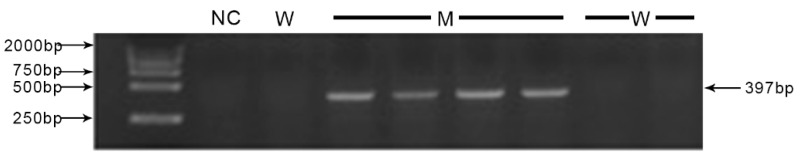
Genotype detection of *MSTN*-mutant progeny. Genomic DNA was extracted from newborn piglets and then was subjected to PCR. The PCR products were used for to detect ZFN-targeted region of *MSTN* gene to distinguish *MSTN*^+/+^ and *MSTN*^+/−^ pigs. NC: H_2_O; W: Wild-type, *MSTN*^+/+^ pig; M: Mutation, *MSTN*^+/−^ pig.

**Figure 2 animals-10-00932-f002:**
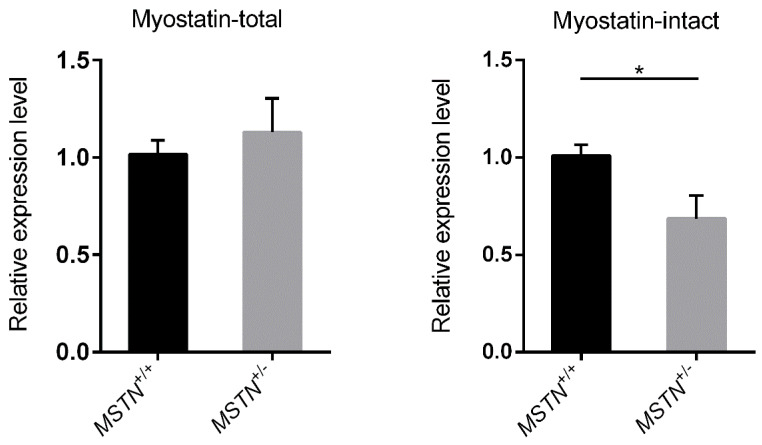
Real time quantitative PCR results of *MSTN*. Total RNAs were isolated from LMA of *MSTN*^+/+^ and *MSTN*^+/−^ pigs. The expression levels were analyzed using the ∆∆Ct method and normalized against *GAPDH*. Each sample was run in triplicate. *: *p* ≤ 0.05.

**Table 1 animals-10-00932-t001:** The composition and nutrient level of the basal diet.

Item	20–50 kg	50 kg-Slaughter
Ingredients (g/kg)		
Corn	715	787
Soybean meal	250	180
Dicalcium phosphate	11	10
Limestone	8	8
Lysine-HCl	2	1
Salt	4	4
Premix ^1^	10	10
Total	1000	1000
Calculated nutrient levels		
Digestible energy (MJ/kg)	13.77	13.80
Crude protein (%)	17.27	14.80
Calcium (%)	0.71	0.66
Total phosphorus (%)	0.60	0.55
Lysine (%)	1.02	0.80

^1^ The premix provided per kilogram of diet: 100 mg of Fe, 100 mg of Zn, 30 mg of Mn, 10 mg of Cu, 0.3 mg of Se, 0.5 mg of I, 2758 μg of vitamin A, 25 μg of vitamin D_3_, 20 mg of vitamin E, 3.0 mg vitamin of K_3_, 2.0 mg of vitamin B_1_, 6.0 mg of vitamin B_2_, 20 mg of vitamin B_3_, 3.0 mg of vitamin B_6_, 30 μg of vitamin B_12_, 8 mg of pantothenic acid, 0.5 mg of folic acid, 300 mg of choline.

**Table 2 animals-10-00932-t002:** Primers sequences used for PCR identification and qRT-PCR analysis.

Gene	Forward Primer	Reverse Primer
*PZFN1/PZFN2*	TACAAGGTATACTGGAATCCGATCT	GCAAAGTAAAAGTATCAAGAGGGTA
*MSTN*-intact	TGAGAATGGTCATGATCTTGCTG	TCCAGTCCCATCCAAAAGCT
*MSTN*-total *GAPDH*	AGTGATGGCTCCTTGGAAGAGTGAAGGTCGGAGTGAACG	TGTAGGAGTCTTGACGGGTCTCGCTCCTGGAAGATGGTG

*PZFN1/PZFN2*: use to detect ZFN-targeted region of *MSTN* gene; *MSTN*-intact = intact *MSTN*, use to detect *MSTN* genetic integrity; *MSTN*-total = total *MSTN*, use to detect total *MSTN*; *GAPDH* = glyceraldehyde-3-phosphate dehydrogenase.

**Table 3 animals-10-00932-t003:** Reproductive related trait data.

BSN	SSN	LS	Alive Litter Size			LW (kg)	BW (kg)
			Male	Female	Subtotal		
452	678	10	4	5	9	9.26	1.03
452	8978	9	3	6	9	10.64	1.18
452	3624	15	6	7	13	13.03	1.00
452	8834	12	6	5	11	10.51	0.96
452	2228	12	5	3	8	9.29	1.16
452	1178	15	8	5	13	13.34	1.03
Total		73	32	31	63		

BSN = boar serial number; SSN = sow serial number; LS = litter size; LW = litter weight of pigs born alive. BW = birth weight of pigs born alive.

**Table 4 animals-10-00932-t004:** Comparative analysis of growth traits.

Item	Barrow		Female		SEM	*p*-Value			
	*MSTN*^+/+^ (*n* = 8)	*MSTN*^+/−^ (*n* = 13)	*MSTN*^+/+^ (*n* = 7)	*MSTN*^+/−^ (*n* = 15)		S	G	S × G	BW
BW (kg)	102.29	101.25	103.57	99.65	3.37	0.173	0.283	0.814	-
ADG (g)	655.11	647.96	674.41	643.27	44.31	0.327	0.460	0.478	0.001
LBF (mm)	22.13	20.83	21.77	19.57	2.71	0.639	0.300	0.662	0.001

BW = final body weight; ADG = average daily gain; LBF = live ultrasound backfat thickness at the 3rd to 4th last rib; S = sex; G = genotype.

**Table 5 animals-10-00932-t005:** Comparative analysis of body size traits.

Item (cm)	Barrow		Female		SEM	*p*-Value			
	*MSTN*^+/+^ (*n* = 8)	*MSTN*^+/−^ (*n* = 13)	*MSTN*^+/+^ (*n* = 7)	*MSTN*^+/−^ (*n* = 15)		S	G	S × G	BW
BL	117.57	114.36	114.25	111.31	4.69	0.021	0.102	0.994	0.002
BH	63.57	61.08	58.87	59.58	2.63	0.001	0.489	0.022	0.001
ChG	110.00	109.00	107.38	106.85	4.43	0.839	0.300	0.147	<0.001
AG	126.43	123.25	126.50	122.77	2.80	0.440	0.088	0.826	<0.001
CaG	16.71	16.25	16.13	16.31	1.12	0.349	0.966	0.457	0.038
HG	82.50	83.83	86.57	84.62	3.94	0.045	0.716	0.184	0.007

BL = body length; BH = body height; ChG = chest girth; AG = abdominal girth; CaG = Cannon Girth; HG = hip girth; BW = body weight; S = sex; G = genotype.

**Table 6 animals-10-00932-t006:** Comparative analysis of carcass traits.

Item	Barrow		Female		SEM	*p*-Value			
	*MSTN*^+/+^ (*n* = 8)	*MSTN*^+/−^ (*n* = 13)	*MSTN*^+/+^ (*n* = 7)	*MSTN*^+/−^ (*n* = 15)		S	G	S × G	BW
Dressing yield(%)	71.65	73.27	72.83	73.82	1.67	0.043	0.001	0.479	0.029
% of carcass									
Lean (%)	46.37	55.41	48.42	54.86	3.97	0.449	<0.001	0.204	0.017
Fat (%)	37.66	29.18	35.98	30.27	4.36	0.718	<0.001	0.195	0.002
Skeleton (%)	7.57	7.39	7.75	7.51	0.90	0.308	0.048	0.894	0.001
Skin (%)	8.40	8.08	7.85	7.37	0.89	0.005	0.016	0.603	0.009
ABT(mm)	33.98	27.69	30.39	26.61	5.74	0.124	0.043	0.352	<0.001
ST(mm)	3.55	3.69	3.88	3.56	0.64	0.600	0.635	0.228	0.249
LMA (cm^2^)	24.31	26.75	25.18	28.91	4.65	0.328	0.003	0.519	<0.001

ABT = average backfat thickness; ST = skin thickness at 6/7 ribs; LMA = longissimus muscle area; BW = body weight; S = sex; G = genotype.

**Table 7 animals-10-00932-t007:** Comparative analysis of meat quality traits.

Item	Barrow		Female		SEM	*p*-Value			
	*MSTN*^+/+^ (*n* = 8)	*MSTN*^+/−^ (*n* = 13)	*MSTN*^+/+^ (*n* = 7)	*MSTN*^+/−^ (*n* = 15)		S	G	S × G	BW
pH_1_ (45min)	6.58	6.65	6.46	6.42	0.08	0.018	0.851	0.588	0.950
pH_2_ (24h)	5.76	5.73	5.73	5.68	0.09	0.339	0.720	0.422	0.692
Color parameters									
Lightness (L*)	41.71	42.13	41.95	42.24	4.36	0.444	0.089	0.229	0.517
Redness (a*)	10.87	11.12	10.72	11.68	0.94	0.272	0.066	0.606	0.936
Yellowness (b*)	4.75	4.34	4.36	4.32	0.51	0.365	0.626	0.157	0.385
Drip loss (%)	1.32	1.08	1.53	1.37	0.24	0.087	0.168	0.084	0.069
IMF (%)	4.59	3.05	4.36	2.66	0.64	0.497	0.010	0.942	0.01

IMF = intramuscular fat; BW = body weight; S = sex; G = genotype.

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
