# Peer review of "Effect of MSTN Mutation on Growth and Carcass Performance in Duroc × Meishan Hybrid Population"

_animals, 2020, doi:10.3390/ani10060932_

Round 1
Reviewer 1 Report
Your manuscript has some merit with regard to the impact of MSTN single allele mutation on carcass measures when assessed across known low carcass quantity breeds. Their are significant challenges in your material that will need to be adjusted prior to publication (in my opinion). Some, but not all specifics are described below. You will want to re-read and review your materials on your own as well to find and fix key components.
The MSTN gene, as it relates to double-muscling, is observed in cattle and most other species as a homozygous condition (-/-) not as you mention or indicate in the heterozygous state (+/-). This needs to be clear in your use of the terms throughout the manuscript. You did not knock down MSTN completely (or you do not see this unless you compare the -/- RNA/DNA expression)
L13 - You should not start a sentence with an acronym.
L14 - you use DM acronym to mean Double Muscle and Duroc Meishan in the same paper. Need to fix this.
L18, L24 - You use the term ...lean meat rate.... in many areas of the manuscript. This is not a term that is common (and likely not correct) so you need to adjust this throughout. I believe you mean lean meat 'yield' or percent lean meat in the carcass. You did not measure a rate (xxx/unit of time)
L29 - 31. You go into detail about the number of pigs born, but no indication of why you only chose 48 of 63 born alive. You need to be transparent in that you did not randomly select these pigs, they are from a finite population and thus a 'captive' set. The key will be that you explain somewhere why you only used 48 and in different proportions. If the intent is to describe reproduction, then show total number of males, females, and genotypes by each sex and perhaps each female to be clear on why and how you identified the final set of animals.
L32 - You indicate the pigs 'could' grow normally. You do not measure could, you measured if they did, under the conditions you have, grow at the same or differing rate.
L34 - suggest you use .. ultrasonic measure of live animal backfat thickness...
What is thinner backfat. You measured fat in many areas, but did not show any of this data in tables or mention in results and or discussion. Be clear and identify which measures you are indicating and why.
Line 45 - What is green crude (define)
Line 50 - What is the word 'autochthonous' I tried to look up this word and could not find it.
L71 - Be sure you define MSTN, in its' mutant form only, reduces fat deposition.
L74 Belgian Blue and Piedmontese?? I do not know of or could find a Pyrmont breed of cattle.
L82 - Here you mention directly the double muscling impact of the knockout (which are -/-) and your study clearly only looks at the heterozygote. I follow your intent, but you need to be clear.
L84 Define SCNT at some point. Not intuitive as currently written.
L88 - How do you know the impacts in this study are not a function of other alleles from the Duroc terminal sire? You do not acknowledge that there are background genes that also influence the traits.
L99 - appears you used pigs near 22 kg at 60 d and this is how you are defining random selection. Try to reword for clarity that you did not/were not able to keep numbers equal across genotypes because you only used a portion that met some defining criteria (this may bias you outcomes later related to growth rate differences as you have pre-selected the study).
L100 - 101 - Ad libitum access is not consistent with feeding twice per day. Was enough provided each time to maintain ad libitum access or were the pigs actually limited between feedings. I think you only need ad libitum access if that is what truly happened.
L111-139 - This portion, the key traits you are reporting on, needs some clarification. The information is not as clearly written and logical in transitions between traits as I would prefer.
For example, last 3rd to 4th rib, do you mean .. third to 4th last rib?
You indicate pigs were weighed at the abattoir in one sentence then mention you transported pigs and rested them prior to harvest. These are contradictory, if the pigs were weighed 24 hours prior to harvest in an abattoir then they were off feed for 24 hours, but this would not correspond with 2hr rest and 1 hr transport.
You mention organs were weighed, but no data shown (either remove wording or at least summarize the outcomes if you mention the measurements).
You also mention many areas of backfat measurement and do not report in the manuscript tables or discussion. Remember your intent for the study was to show carcass differences, but you do not extend your results to show all of the important traits you mention.
Consistent use of L. dorsi or longissimus dorsi (you use both and not consistent)
Not sure what a 108 standard observer refers to? are you talking degrees of angle? Have not seen this terminology. Do you mean 2.5cm diameter?
L184. Were the pigs fed by sex? Were the males castrated? Were sex effects included in the model (known source of variation that must be accounted for in the analyses). A major concern regarding interpretation of your findings.
Figure 1 - This is a difficult to interpret figure. Why are the W designations on both sides of the M designations.
L141 - Can't start a sentence with a number (8).
L140 to 141- May not be needed in this document. What does it add, can it be reduced to the key components.
L151- Subheading - Do you mean 'Genotype Assessments' or as you state it 'Phenotypes measurements'
Table 3 - This table information is not necessary. Would rather see number per sow used in the actual study of carcass composition.
Standard Error of Means: These are obviously pooled since the SEMs are different based on number of animals in each classification. So indicate pooled SEM (or whatever they actually are)
Results in general: Very non-scientific wording, too casual. Appears to be different authors throughout. Try to blend better.
L236 L* is not considered luminosity. L* is objective color on a 0 to 100 scale.
L251 - Litter sizes are normal for industry, so not sure why you think pigs were small from a meishan mother
Qualify all of your discussion regarding MSTN knockouts as to their true genotype. It is not clear if you have comparisons between heterozygotes and homozygotes. This is important in the context of study.
L292 - What are wide type pigs? Do you mean wild-type?
Reviewer 2 Report
Animals - animals-764952
This manuscript is well-written with a relevant scope and a sound experimental setup. Assessing a new cross-breed of indigenous Chinese Meishan and the high performing Duroc aiming for MSTN+/- was successfully bread and investigation of production parameters as well as quality of the meat is very relevant. The theoretic reasoning for the genetic aim is well-described, relevant parameters are included, and data is analysed and presented in a logical way leading to sound conclusions. This may not be ground-breaking but sound research and a relevant extension on existing knowledge.
Only minor suggestions:
Line 113: change “measure” to “measured”
Line 114: is “h” missing after “24”?
Line 120. “And” could be changed to “They”
Lines 135-136: was the meet left to bloom for a specified period? Please include this period – or include a reasoning for not blooming if this was the case.
Line 141: insert “a” before “local”
Line 149: insert “and” before “litter
Line 200: could the sentence be improved by adding “in” and “it” i.e.: “…while in wild-type offspring it could not be detected.” ?
Line 216: change “showed” to “shown”
Line 244: change “And” to “Also”
Line 251: change “to a small extent” to “to some extent” and change “All this” to “All these”
Lines 258-260: this sentence is a bit difficult to follow. Could this be re-worded?
Lines 270-272: change “12% and 4% than wide type pigs, respectively” to “12% and 4%, respectively, compared to wild type pigs”
Line 271: consistent with what? Consistent whit those found in the present study?
Line 285: should “one result” be changed to “a consequence”?
Line 292: change “”And” to “However, “
Line 289: delete “high” before “pH” and “were” before “within” i.e. “had pH (45 min) values, within normal scopes”
Lines 305-309: I suggest that you delete the mention of fiber types as you have not analysed fiber types and have no reason to suggest that the distribution is different (the color is not different i.e. the a*-value is the same)
Lines 39 and 319: I am not sure if it is correct to call this a local fat-type indigenous pig breed or t should be called a “cross-breed”?
Round 2
Reviewer 1 Report
Effect of MSTN mutation on growth and carcass 2 performance in Duroc x Meishan hybrid population
General Comments:
The authors have made significant improvement to the manuscript when compared to the previous version. The information relative to the mutation and the processes used in the study are clearer, but will still need some additional adjustment to meet expectations. Terminology, which leads to grammar and sentences structure issues, used by the authors is still a challenge and will need to be clarified. Specific response are listed below.
L13 – Indicate that a single allele mutation of MSTN is sufficient to result in a double muscle phenotype (This is what you are proposing in your study, ie. the heterozygote and homozygote mutant both show double muscling) This shows the inheritance of the double muscling condition as being dominant. Is this correct? Your reader needs to know.
L16. You use the term low feed conversion. I suggest you use the term ‘poor’ or ‘poorer’ feed conversion throughout to alleviate any concern related to FCR as Feed:Gain ratio being lower and being better.
L20: Do you propose knocking out the MSTN gene in local breeds? Or are you proposing using the local breeds as they are and capturing the value they offer in regard to reproduction (litter size, etc) and meat quality? May wish to clarify.
L25. You did design the set out ‘to improve’ …. Rather, I believe you might say. To evaluate lean meat yield of Meishan …………
L26. You inseminated 8 it appears, but only 6 had litters. Tell your readers that you used progeny from 6 Meishan sows successfully mated to the Duroc boar. You muddy the interpretation when you insinuate the pigs used came from 8 sows.
L28. ….ensure that semen from MSTN knockout boars did not affect…….
L30. Male to female ratio was close to 1:1
L30-31 ……..only 48 pigs were utilized, including …..
L32. No differences were observed between MSTN+/- and MSTN+/+ genotypes for live animal measures including average daily gain, body dimensions, or ultrasonic measurement of fat thickness when pigs were harvested at a standard weight of 100 kg. NOTE: Your chosen end weight is not a trait in your study per se. Your words indicate you stopped the trial when pigs reached 100 kg, not that you measured the time (days) it took for each of the genotypes to reach 100 kg. There is a difference in design and interpretation of the outcomes. If you measured Days to 100 kg (a true trait of potential interest) you can report this. Otherwise you have comparisons that reflect a target end weight of 100 kg.
L35 to 38. The plural and singular forms of words need to be addressed. For example are you sure you want the word ‘percentages’ or is it more likely ‘percentage’
L38 ….. we got…. Not a scientific way to indicate your outcome. How about. In conclusion, the production of MSTN+/- mutant progeny from Meishan females resulted in improved carcass composition, providing a ……..
L48 …poor feed ….
L54. Meishan pigs are a ---
L55 and throughout the manuscript. Fix sentences that start with the word And. Grammar rules suggest you do not start sentences with the word ‘And’. There are many instances of this in the manuscript that need to be fixed.
L60. …percentage of lean carcass weight…. A percentage is a percentage, weight is weight. They are not the same. I think you mean ….. percent lean in the carcass… or ….percent carcass lean…..
L62-63. For the past many years…. This is not a correct statement and the wording of the sentence as written makes this an incomplete sentence. Restructure the sentence for clarity.
L64. What do you mean by ‘Nowadays’? This is not a scientific word and has no context as used. Reword.
L67-74. Please indicate the inheritance of the MSTN gene and the genotype effects. Is this a dominant condition, and which alleles are considered normal vs mutant. You have a need to be certain your reader knows what is implied or known about the genotypes +/+. +/-, and -/-
L74. ….. the MSTN mutation. Add word the
L75. Thus, …. Consider. Within livestock industries there has been great attention focused on MSTN inhibition…..
L77 …. Belgian Blue and Piedmontese breeds of cattle. Muscle mass and lean meat yield increased greatly in these breeds due to the mutation, resulting in what is termed a ‘double muscle’ phenotype.
L82. Question that goes back to a comment earlier and other instances in your introduction. Are you wanting to develop new breeds? Or are you wanting to produce progeny from local breeds that have the MSTN mutation so that you can gain all of the reproduction opportunities and observe the increase in carcass yield. These are two different goals and objectives and both have merit, but your message as currently written is not clear from one location in the manuscript to another.
L87. This paragraph helps a bit to decipher the intent of your manuscript with regard to potential negative impacts of the mutation on reproduction traits. Take a close look at the sentence starting with L87 Considering… This is not a well-written sentence and can be reworded to get the many concepts you are trying to indicate presented in a more precise manner.
L92. Would the words. …… growth and carcass composition and quality in …..
L122 - ….. included body length…..
L123 ….caliper.
L124. Backfat depth was measured using ultrasound (Mylab….. ) on the left side of the live pig at the 3rd to 4th last rib location.
L126-128. Pigs were removed from feed the night before slaughter, transported in the morning 1 h to the abattoir, and allowed to rest for 2 h with access to water prior to slaughter. Pigs were electrically stunned (xxx)..
This appears to be the way pigs were handled based on the overnight fast described in Line 137. OR, you have two different scenarios being presented and one of them is not correct.
L130. The left carcass side was….. (Assuming at 24 h post harvest minimum on the chilled carcass, not as a hot carcass. Is this the situation or was the left side of the carcasses fabricated hot?
L132. The organs removed (do not need this sentence. You mention evisceration in Line 129)
L179 Statistical analyses. This is an important part of your study and one that needs to be explained correctly for interpretation of the results.
Some important comments and questions. First, in later paragraphs (L189, L256-263, Etc.) you mention sex effects, yet you do not show how many males (boars) and females (gilts, not sows as they have not had a litter) were in the genotyped set of pigs evaluated. You mention in paragraph starting at L256 that sex effects are not found in the literature. This is statement is simply false. The effect of intact males vs. castrated vs. intact females are identified and present in literally 100s of manuscripts related to swine research. Your assumption and failure to account for sex effects on growth traits, carcass composition traits, and meat quality traits, will not allow publication of your data. These effects must be included in your statistical models for all traits (sex is a fixed dependent variable), testing genotype by sex interactions first and assuring that genotype effects are reported after accounting for sex and or sex x genotype effects. In addition, because males grow faster than females (based on literature expectations), it is not clear if boars were harvested on different days (assuming you have indicated all were harvested at ~100 kg) than gilts. If this is the case, day of harvest would need to be a random effect in your carcass quality measurements as day effects are also well studied and known to occur and which can bias genotype effects with sex of pig. In addition, weights of carcass components, backfat thickness measurements, loin area, and any other measurements of weight or dimension will need to have a covariate included in the model to account for known variation in live or carcass weight at the time of harvest (standardize to a common weight basis). You do not do this is you are calculating percent lean directly as weight is already in the model.
The explanation of sex effects in Lines 256 – 261 is not a good interpretation of science nor justifiable for not including sex in the model. Even in studies where sexes are fed to meet nutritional requirements for the given sex, there are differences between the sexes. We are not able to account for sex effects by feeding, management nor end weights that approach 100kg or greater. Please look further into the research.
L222. …live ultrasound backfat thickness at the 3rd to 4th last rib.
L229 … higher percentage of lean carcass weight ( ). Should be greater percentage of lean in the carcass, or greater carcass percent lean. You can’t have both weight and percentage describing the same trait, it is one or the other.
L233. Are you sure the 5.77 vs 5.53 pH is not significant. This seems to be a very large difference numerically given that this is a log scale. Check the stats.
L249. Did you inseminate 6 or eight Meishan sows? Why does the number of sows change from earlier in the manuscript? You need to be consistent in your approach. You have continually indicated that reproduction was not a primary focus of the study, yet if you are being transparent in your results is 6 of 8 mated producing a litter a good reproductive outcome? Is litter size truly equal to non-mutant pigs? The point of this discussion is that you re-word the same information in your abstract, compared to your introduction, compared to materials and methods, compared to your results, and compared to your discussion sections and it has created confusion with your true intent. State plainly the approach in the materials and methods section and use this standard throughout. If you collectively wish to state no differences in reproduction that is fine, but please be consistent in the message vs. how you now have this written which is both confusing and takes up excessive space in your manuscript.
L266. What are … “Further body size measurements shown’….. This is not a functional sentence. Re-word.
L269. What are … ‘unobvious traits’ Not clear as to intent and reason for including this type of statement
L271. What are …’certain effects’. Your study did not address a comparison of live and carcass data for backfat. You did not measure the live animal in the same anatomical locations as the carcass. Third to fourth last rib in modern pigs is rib numbers 14, 15, 16, 17. This is not the tenth rib (where LMA was measured, was Backfat also measured on the split carcass at the tenth rib?). Was the ultrasound technician trained to measure BF in pigs?
L280 … without changing the proportion of Meishan pigs. What does this sentence indicate? Do you suggest that if you used a double mutant sire (always transferring the – allele) that when mated to Meishan all pigs would be leaner as they would all inherit the mutant allele) and the industry would be able use purebred Meishan in breeding programs without the need for percentage inclusions of Meishan that are now being incorporated?
Your group continually goes back to prolificacy of the Meishan, a known and published phenomenon, as the basis for the paper. So, just go ahead and state it more clearly. You group is attempting to improve carcass composition while maintaining prolificacy (which many say is better than current genetic resources used commercially).
Round 3
Reviewer 1 Report
Much improved over previous versions of the manuscript. A few notes that wil need your attention.
First, as I have read the manuscript from first to current versions, it is readily apparent that your group is very well versed in fundamental, basic biological processes and for that you are commended. The primary challenges you still encounter are related to less familiarity with the swine and meat industries, which show up in wording choices and concept discussion. Therefore, I think it will be important to continue to have someone directly involved in research at the applied swine breeding level review the final version to ensure the messages you intend to show are presented in a format that will be easily understood and clear to your audience.
The following recommendations are, in my review, important enough thtat you will want to clarify before the final version is submitted.
Line 34. ...average daily gain....... (remove s from gains).
Line 37 and throughout. You use the words ... loin eye area... in some sections of the manuscript but also use L. dorsi (140, and 142, 143-44, 158.) You have not made the connection of loin eye area (LEA) with L. dori in your wording. Thus, perhaps you need to start with a definition of your measurement at line 140. Longissimus muscle area (LMA) was determined..... (you do not need dorsi as all of the longissimus is dorsal on the carcass. You could but do not need to indicated longissimus thoracis (thoracic area of the Longissimus is where you are measuring area)). If you make changes at L140 you can use LMA from that point forward. Please verify consistent use of the terminology across all sections of the document.
L137. Remove the sentence - The organs were removed. This was already stated as evisceration in Line 134.
L144. Please verify that color parameters were measured at 45 minutes. If the measurements were taken at 45 minutes then the reference to NPPC standards (line 319), which are color standards for pork at 24 to 48 hours post death (not 45 minutes) are not correctly interpreted. IF you have 45 minute (vs 24 hour when the second pH measurement was taken) your interpretation of color against NPPC standards would be expected to be have darker L* (lesser number) because at 45 minutes post death the carcass is still at a very high pH (not yet reached rigor, nor dropped much from live pH and the pork should be dark in color). The NPPC standards reflect color at 24 to 48 hours post death. Therefore, you must clarify when color measurement took place, and adjust wording to reflect correctly any references to literature that reflect measurements at 24 hours post death. For example, reading Lines 318 to 319, you mention PSE, DFD, and Acid Meat as examples of meat quality defects, but each of these parameters are determined by color (as on variable) measured at 24 hours after death and not at 45 minutes after death.
L202. Please fix this sentence. Due to issues with disease in some of the available pigs, 43 pigs were utilized in the growth and carcass composition and quality assessment components of the study, including 21 castrates and 22 females. Another common term for castrates is barrow. Use either, but the pigs you are assessing are castrated.
L205 (13 castrates,..
L206 (8 castrates, ..
Tables 4, 5, 6, 7 . Replace male with castrate (or barrow)
L315 - Try not to start sentence with an acronym. ... The pH of pork is a major parameter that defines meat quality ......
One final thing. Please review for plural vs. singluar forms of verbs and nouns.
